# The effect of software and hardware version on Apple Watch activity measurement: A secondary analysis of the COVFIT retrospective cohort study

**Shelby L. Sturrock**[1]*, **Rahim Moineddin**[2], **Dionne Gesink**[1], **Sarah Woodruff**[3], **Daniel Fuller**[4]

**1** Division of Epidemiology, Dalla Lana School of Public Health, University of Toronto, Toronto, Canada, **2** Department of Family and Community Medicine, University of Toronto, Toronto, Canada, **3** Department of Kinesiology, University of Windsor, Windsor, Canada, **4** Department of Community Health and Epidemiology, College of Medicine, University of Saskatchewan, Saskatoon, Canada

* shelby.sturrock@mail.utoronto.ca

## Abstract

The objective of this study was to estimate the impact of software and hardware version on Apple Watch activity measurement using data from the COVFIT retrospective cohort study. We estimated the impact of software and hardware versions on activity measurement by comparing daily active calories and daily exercise minutes in the 7 days before and 7 days after upgrading from watchOS 5 to 6, 6 to 7, 7 to 8, 8 to 9 or between two hardware versions. For each transition, we fit mixed effect negative binomial regression models to estimate the effect of the upgrade on daily (a) exercise minutes and (b) active calories, overall and stratified by sex, with and without adjusting for weekday. We also calculated and plotted the mean person-level change in average activity levels between the two weeks. As a control, we repeated the entire analysis comparing activity data two weeks before vs. one week before each upgrade. 253 participants contributed data about at least one transition (software = 250, hardware = 74). Hardware upgrades were not associated with either outcome; however, some software upgrades were. Upgrading from watchOS 7 to 8 was associated with a large, statistically significant increase in daily exercise minutes (unadjusted rate ratio (RR) = 1.13, 95% CI: 1.06, 1.20). WatchOS 6 to 7 and 8 to 9 transitions were associated with statistically significant decreases in daily exercise minutes (6 to 7: unadjusted RR = 0.92, 95% CI: 0.86, 0.99; 8 to 9: unadjusted RR = 0.91, 95% CI: 0.86, 0.96) and active calories (6 to 7: RR = 0.96, 95% CI: 0.94, 0.99); 8 to 9: RR = 0.97, 95% CI: 0.94, 0.99). There was no significant change in either outcome during in the two-week control period for most transitions. Differences in software version over time or between people may confound physical activity analyses using Apple Watch data.

**Data availability statement:** The study involved primary data collection did not obtain consent from participants to share their data publicly. Participants were informed that "Results of the study will be presented in presentations and publications in aggregate form to maintain confidentiality and anonymity." As such, individual-level data cannot be shared publicly. Questions about access to these data can be directed to the University of Toronto Health Sciences REB (ethics.review@utoronto.ca).

**Funding:** The author(s) received no specific funding for this work.

**Competing interests:** The authors have declared that no competing interests exist.

## Author summary

Researchers are increasingly using data from participant's personal wearable devices, like Apple Watch, to study physical activity over time and between people. These people may all be using different hardware and/or software versions, which may impact how physical activity is measured. If this is the case, comparisons of physical activity levels over time, or between people, may represent the combined effects of differences in behaviour, as well as differences in how that behaviour is being measured. Using data collected for the COVFIT study, we compared daily active calories and daily exercise minutes in the week before and the week after our participants upgraded between two major software versions (e.g., watchOS 5 to watchOS 6), or switched to a different hardware version. We found large changes in daily exercise minutes and active calories in the week after compared to the week before some, but not all, major software upgrades. In future, researchers should measure and account for hardware and software version when either may differ between participants and/or over time.

## Introduction

Consumer wearable devices are a new and promising tool for measuring physical activity in research studies. Smartwatches and fitness trackers contain sensors that collect near-continuous data about the user's health and movement [1]. In the past, such data could only be collected prospectively, by providing participants with a research-grade accelerometer, typically for seven to 10 days [2]. This process is burdensome for participants and logistically difficult for researchers, making it unsustainable for long-term data collection. As a result, little is known about longitudinal trends in individual-level physical activity.

Instead, researchers can now leverage data collected by participant-owned wearable devices. These data are expected to be more complete and span longer periods than data collected specifically for a research study. The devices are also ubiquitous; over one-third of American adults reported using a wearable device to track their health and fitness as far back as 2019/2020 [3].

Importantly, little is known about how consumer wearables measure, process and summarize activity metrics. Raw data are often not available, and manufacturers do not indicate how summary metrics such as "minutes of exercise" (Apple), "active calories" (Apple) or "active zone minutes" (Fitbit) are calculated. Many studies address these limitations by restricting to one device type. However, physical activity measurement may also differ between hardware and software versions of the same device, posing a significant threat to the validity of research leveraging consumer wearables. New hardware versions may contain new and/or improved sensors, while software upgrades could alter activity processing algorithms. This means that the accuracy of physical activity data from wearable devices may vary between people and/or over time depending on software or hardware version [4]. Furthermore, comparisons of physical activity levels over time or between individuals may reflect the combined effects of differences in behaviour and differences in measurement [4], due to differences in software or hardware versions. Identifying and quantifying the effect of software and hardware changes is needed to improve the validity of longitudinal research using commercial wearable devices.

While researchers have noted the potential impact of software or hardware version on activity measurement – and thus on research using commercial wearable devices [4] – no studies have attempted to quantify this effect. Studies of the "firmware effect" among

research-grade accelerometers have found statistically significant differences in activity measurement between research-grade accelerometers (Actigraph) running different versions of firmware [5]. In 2016, Actigraph identified a bug in their firmware that led to changes in activity measurement relative to previous versions [6], highlighting the role that software and/ or firmware plays in activity processing by wearables. Studies leveraging data from commercial wearables have speculated that different software and/or hardware versions across devices or over time may impact their findings, but have not estimated this effect [7].

The Apple Watch is the world's most popular wearable device. Apple releases new hardware and major software versions every Fall, often introducing new sensors and changes to system functionality that may alter activity measurement. The objective of the current study is to estimate the impact of software and hardware version on the measurement of daily activity metrics by Apple Watch.

## Methods

### Study design and data collection

This study is a secondary analysis of data from the COVFIT study, a retrospective cohort study leveraging routinely collected Apple Watch data to study longitudinal trends in physical activity before and throughout the COVID-19 pandemic. To be eligible, individuals needed to own an Apple Watch, be 18 years of age or older and live in Canada at the time of participation. Recruitment was conducted via word of mouth and social media between August 2022 and March 2023.

Data were collected using the COVFIT study app, which was developed by the University of Toronto Mobile Application Development Lab using Apple's ResearchKit framework [8]. It was available on the Canadian Apple App Store from August 2022 until May 2023. Participants were asked to install the COVFIT study app on their iPhone, provide written consent to participate, provide read-only access to select historic activity data and complete a survey. The app accessed activity data stored in HealthKit, including data recorded by current and previous hardware associated with the user's Apple ID. The following metrics were extracted from January 1, 2019 (or as early as available, whichever was more recent) until the date of participation: daily exercise minutes, daily active calories, minute-level heart rate, and the hardware and software version that recorded each heart rate measurement.

### Patient and public involvement

**The public was not involved in this research.**

**Ethics statement.** The COVFIT study received ethics approval from the University of Toronto Health Sciences Research Ethics Board (protocol #39612).

### Outcomes

The primary outcome was daily minutes of exercise as measured by the participant's Apple Watch. This is an estimate of moderate-to-vigorous physical activity and is defined as "movement at or above the pace of a brisk walk" [9]. A secondary outcome was daily active calories, which measures the "energy that the user burned due to physical activity and exercise", excluding resting energy burned during the sample's duration [10].

### Exposures

The exposures of interest were day-level software and hardware versions.

A new major version or "upgrade" of the Apple Watch operating system (watchOS) ("software") is released every September. These are planned well in advance and make important

changes to system functionality. Four major software versions were released during the study period: watchOS 6 (2019), 7 (2020), 8 (2021) and 9 (2022). Apple also released minor updates throughout the year to fix bugs and/or improve security. We assumed that activity measurement would only change after upgrading between major software versions (e.g., watchOS 5 to 6), and not after installing a minor update (e.g., watchOS 5.1 to 5.2). As such, software version was defined as the *major software version* running on the user's device on a given date, based on the minute-level "sourceVersion" variable. This was then aggregated to the day-level using mode.

New Apple Watch devices are also released each Fall. Most new hardware versions (e.g., Apple Watch Series 8 or Apple Watch SE) are available in different sizes and with/without cellular connectivity. We assumed that activity measurement could vary between new hardware versions (e.g., Series 7 to 8), but not between different models of the same hardware (e.g., Series 8 with vs. without cellular). As such, hardware version was defined as the commercial name of the Apple Watch that the participant used on a given date (e.g., Series 8), regardless of size or cellular connectivity. This was mapped from the internal product name recorded at each heart rate measurement ("productType") using an index of Apple Watch versions [11], and aggregated to the day-level using mode.

## Data processing

Person-days were excluded if daily exercise minutes or daily active calories were in the 99.9th percentile for the participant's sex assigned at birth ("sex"), or if total wear time was under 600 minutes per day [12–16]. We estimated wear time using heart rate data, which is only recorded while the Apple Watch is on the user's wrist. Measurements are taken continuously during workouts and less frequently otherwise [17]. As of 2015, passive measurements were attempted every 10 minutes but were not recorded if the user was in motion [18], and a recent study found that Apple Watch records a heart rate measurement every 5.62 minutes on average [19]. Intervals of 40 minutes or longer without a heart rate measurement were classified as non-wear segments. Daily non-wear time was calculated for each participant by summing the length of all non-wear segments on that date.

We estimated the effect of each transition of interest (i.e., from watchOS 5 to 6, 6 to 7, 7 to 8, 8 to 9 and between any two hardware versions) by comparing daily active calories and exercise minutes in the 7 days before and 7 days after (inclusive of) the upgrade. This 14-day "transition period" was unique to each participant and was centered around their transition date, defined as the first day when most of their heart rate measurements were recorded on the new software or hardware version (Fig 1, Panels A and B). To estimate normal week-to-week variability, we also conducted a control analysis for each transition, comparing activity data from 15-8 days before and 7-1 day before that upgrade ("control transition period") (Fig 1, Panel C). In both cases, a 14-day window was chosen to balance day-of-the-week effects between the pre- and post-periods and reduce the risk that the transition effect was confounded by underlying trends and/or seasonality, particularly if most participants upgrade at the same time of year.

To isolate the effect of each major software version, only transitions between consecutive major software versions were retained for analysis (i.e., watchOS 5 to 6, 6 to 7, 7 to 8 and 8 to 9). Transitions were also excluded if major software version changed more than once during the 14-day period (e.g., during the 7 days before or after the transition of interest), if hardware version changed at any point during the 14-day period, or the user had no activity data in one or both weeks.

Very few people transitioned between consecutive hardware versions, so the effect of each could not be estimated separately. Any transition between two hardware versions was

**Fig 1. An infographic outlining the data used for each analysis.** First, we identify the transition date for each participant. We then center participants on their transition date and compare activity data from the week before and after (inclusive of) the transition date ("transition period"). As a control, the analysis was repeated on two weeks of data from just before the upgrade ("control transition period").

considered for analysis. Like the software analysis, transitions were excluded if the user had no activity data in one or both weeks, hardware version changed in one or both weeks, or software version changed at any point in the 14-day period.

## Statistical analysis

The primary and control analyses were restricted to participants with activity data on all 14 days of the transition period and control transition period, respectively.

**Descriptive analyses.** We first calculated participant-level changes in mean daily exercise minutes and mean daily active calories from the week before to the week after each of the five transitions. Since activity processing algorithms may be sex-specific and thus software and/ or hardware upgrades could impact male and female users differently, the mean and standard deviation of the change in daily exercise minutes and active calories were plotted for each transition, overall and stratified by sex. As a control, this was repeated comparing data from 15-8 days before and 7-1 days before each given transition, where no software or hardware upgrades occurred.

**Regression analyses.** We also ran a model-based analysis, whereby separate mixed effects negative binomial regression models with a random intercept per participant were fit to daily activity data from the 7 days before and 7 days after each of the five transitions (four software, one hardware), overall and stratified by participant sex. Each model contained a binary variable classifying days as before (0) or after (1) the transition of interest, with and without adjusting for day of the week. This analysis was conducted separately for each outcome (daily exercise minutes and daily active calories) and was repeated using data from the control transition period.

**Sensitivity analyses.** The descriptive analysis was re-run on participants with at least 1, 4, 5, and 6 days of activity data in both the pre- and post-upgrade periods. These results were plotted and compared to the primary analysis. Mixed effects negative binomial regression models were re-run on all participants with at least one day of activity data in both the week before and week after the transition of interest.

**Equity, diversity and inclusion.** The author group is an interdisciplinary team of male and female researchers based in Ontario and Saskatchewan. Our study population included male and female adults of varying ages, races and income levels. The analysis was stratified by sex, as we hypothesized that the effect of software version could vary between male and female participants but not other demographic factors.

## Results

### Sample characteristics

A total of 383 Apple Watch users participated in the COVFIT study, including 379 that shared their activity and heart rate data. On average, the length of time between January 1, 2019 (or first date of activity data, whichever was later) and enrollment was 1125 days (median 1335). Activity data were available for 82% of days on average (median 94%) but varied greatly between participants (minimum 2%; maximum 100%). Male participants tended to contribute data for longer periods of time (median 1454 days) and had activity data for a higher percent of days (median 97%) compared to female participants (median 1135 days; median 89%).

Nearly all participants upgraded major software versions at least once (n = 359; 95%), and 70% (n = 268) upgraded to all major versions released during their respective study periods. On average, participants upgraded to a new major software version every 400 days, and within 25 to 31 days of release. Less than half (n = 183; 48%) of participants switched hardware versions during their study period, and a small number of participants (n = 25) appeared to frequently alternate between two devices.

In total, 1125 relevant software transitions were recorded among 359 participants. Of these, 233 transitions were excluded because major software version changed more than once during the 14-day transition period (n = 103), hardware version changed during the 14-day transition period (n = 68), or there was no activity data in either the week before or after the transition date (n = 62). Of the remaining 892 transitions, 532 had activity data for all 14 days of the transition period and were included in the primary analysis (Table 1). Demographic characteristics were similar across transitions (Table 1). Characteristics of the full sample are described in S1 Table.

Of 1732 hardware transitions, 147 met inclusion criteria and 97 were included in the primary analysis (Table 1). Notably, most excluded hardware transitions (n = 1467) were contributed by a small number of participants (n = 25).

Overall, 253 participants contributed data about at least one software (n = 250) or hardware transition (n = 74) (Table 1). Almost 75% of these participants had a household income over $100,000, half were aged 30 to 49 years of age, 60% were male and nearly 80% had at

**Table 1. Participant characteristics, overall and for each transition, of primary analytic sample (activity data for all 14 days of the respective transition period).**

| | Overall n = 253 | Software transitions | | | | Change in hardware n = 97[*] |
|---|---|---|---|---|---|---|
| | | 5 to 6 n = 93[*] | 6 to 7 n = 114[*] | 7 to 8 n = 166[*] | 8 to 9 n = 159[*] | |
| **Sex = Male (%)** | 155 (61.3) | 62 (66.7) | 80 (70.2) | 104 (62.7) | 106 (66.7) | 78 (80.4) |
| **Age (%)** | | | | | | |
| 18-29 | 27 (10.7) | 10 (10.8) | 15 (13.2) | 14 (8.4) | 13 (8.2) | 7 (7.2) |
| 30-49 | 127 (50.2) | 44 (47.3) | 58 (50.9) | 84 (50.6) | 79 (49.7) | 58 (59.8) |
| 50-69 | 89 (35.2) | 39 (41.9) | 37 (32.5) | 63 (38.0) | 59 (37.1) | 30 (30.9) |
| 70+ | 10 (4.0) | 0 (0.0) | 4 (3.5) | 5 (3.0) | 8 (5.0) | 2 (2.1) |
| **Household income, 2019 (%)** | | | | | | |
| Under $50,000 | 16 (6.3) | 7 (7.5) | 8 (7.0) | 8 (4.8) | 10 (6.3) | 10 (10.3) |
| $50,000 to $99,999 | 48 (19.0) | 15 (16.1) | 22 (19.3) | 32 (19.3) | 32 (20.1) | 15 (15.5) |
| $100,000 to $149,999 | 62 (24.5) | 20 (21.5) | 20 (17.5) | 36 (21.7) | 41 (25.8) | 25 (25.8) |
| $150,000 to $199,999 | 46 (18.2) | 16 (17.2) | 25 (21.9) | 35 (21.1) | 31 (19.5) | 17 (17.5) |
| $200,000 or more | 72 (28.5) | 33 (35.5) | 36 (31.6) | 52 (31.3) | 37 (23.3) | 29 (29.9) |
| No income | 2 (0.8) | 0 (0.0) | 1 (0.9) | 1 (0.6) | 2 (1.3) | 0 (0.0) |
| Don't know/ prefer not to answer | 7 (2.8) | 2 (2.2) | 2 (1.8) | 2 (1.2) | 6 (3.8) | 1 (1.0) |
| **Education (%)** | | | | | | |
| Graduate degree | 95 (37.5) | 36 (38.7) | 47 (41.2) | 67 (40.4) | 56 (35.2) | 29 (29.9) |
| Undergraduate degree | 103 (40.7) | 37 (39.8) | 41 (36.0) | 65 (39.2) | 70 (44.0) | 47 (48.5) |
| Trade or technical school | 26 (10.3) | 9 (9.7) | 12 (10.5) | 16 (9.6) | 12 (7.5) | 9 (9.3) |
| High school or less | 24 (9.5) | 9 (9.7) | 13 (11.4) | 15 (9.0) | 16 (10.1) | 11 (11.3) |
| Don't know/ prefer not to answer | 5 (2.0) | 2 (2.2) | 1 (0.9) | 3 (1.8) | 5 (3.1) | 1 (1.0) |
| **Baseline daily physical activity level (week before the transition) (mean (SD))** | | | | | | |
| Exercise minutes, overall | 48.97 (34.05) | 42.80 (32.83) | 48.62 (29.45) | 45.55 (31.80) | 54.76 (36.95) | 51.62 (37.77) |
| Exercise minutes, female | 48.15 (29.68) | 40.59 (33.11) | 49.48 (23.74) | 42.82 (26.89) | 58.11 (31.90) | 47.68 (30.77) |
| Exercise minutes, male | 49.34 (35.92) | 43.91 (32.90) | 48.25 (31.70) | 47.18 (34.42) | 53.09 (39.27) | 52.58 (39.40) |
| Active calories, overall | 710.43 (250.62) | 741.24 (251.29) | 738.68 (256.40) | 685.16 (248.21) | 681.56 (250.23) | 738.24 (242.74) |
| Active calories, female | 594.31 (193.63) | 614.97 (224.24) | 625.21 (214.75) | 586.12 (182.80) | 576.08 (182.79) | 582.86 (174.36) |
| Active calories, male | 764.17 (255.92) | 804.37 (241.56) | 786.91 (258.60) | 744.21 (263.59) | 734.30 (263.15) | 776.09 (242.80) |

[*]Data is at the transition-level, not the person-level.

least an undergraduate degree (Table 1). The cohort was quite active, recording an average of nearly 50 minutes of exercise per day in the week before each upgrade of interest (Table 1). Male participants achieved higher active calories across all transitions, while daily exercise minutes were similar for male and female participants until the week before the watchOS 6 to 7 transition, when sex-differences emerged.

## Software version

In descriptive analyses, mean daily exercise minutes were, on average, 5.5 minutes higher in the week after vs. the week before upgrading from watchOS 7 to 8 (Fig 2 Panel A; S2 Table). The average change was larger for female (8.7 minutes) than male participants (3.5 minutes). In mixed effects negative binomial regression models, the watchOS 7 to 8 transition was associated with a 13% increase in daily exercise minutes (unadjusted rate ratio (RR) = 1.13, 95% CI: 1.06, 1.19; adjusted RR = 1.13, 95% CI: 1.06, 1.19) (Table 2). The effect was similar in

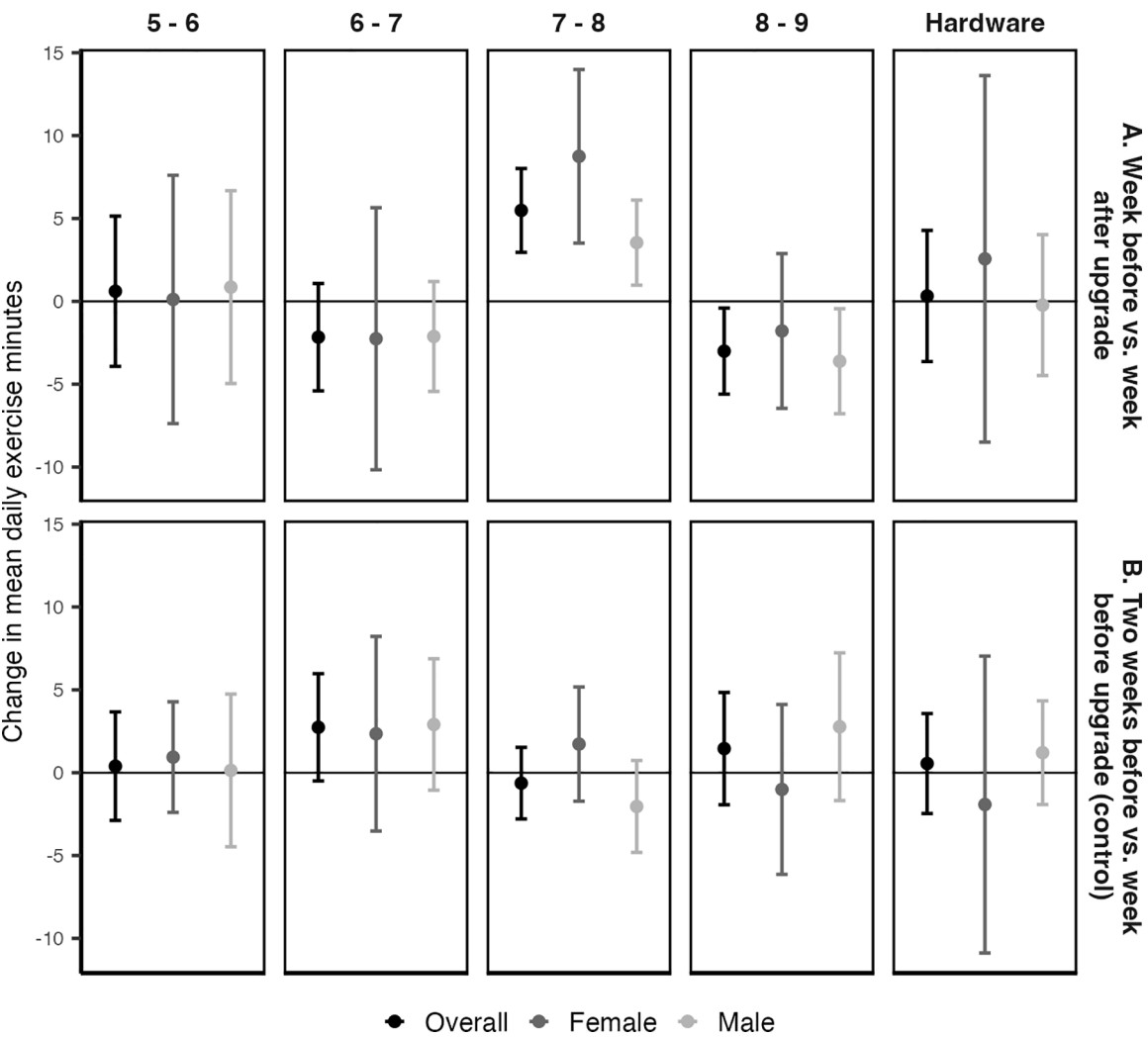

**Fig 2. Average difference in mean daily exercise minutes in the week before and the week after transitioning between the indicated software versions.**

sex-stratified models (Table 2). Importantly, there was no significant change in daily exercise minutes in the control analysis comparing data from two weeks before vs. one week before the watchOS 7 to 8 upgrade (Fig 2 Panel B; Table 2; S2 Table).

The watchOS 6 to 7 and 8 to 9 transitions were both associated with a decrease in daily exercise minutes. In descriptive analyses, mean daily exercise was approximately 2 minutes lower in the week after compared to the week before upgrading from watchOS 6 to 7, and approximately 3 minutes lower after upgrading from watchOS 8 to 9 (Fig 2 Panel A; S2 Table). In negative binomial regression models, the watchOS 6 to 7 transition was associated with an 8% reduction in daily exercise minutes (unadjusted RR = 0.92, 95% CI: 0.86, 0.99; adjusted RR = 0.93, 95% CI: 0.87, 0.99) (Table 2). The effect size was similar in sex-stratified models, though the relationship was not statistically significant among female participants. The watchOS 8 to 9 transition was associated with a similar sized reduction in daily exercise minutes, but the effect was larger among male participants (Male: unadjusted RR = 0.89, 95% CI: 0.83, 0.96; Female: unadjusted RR = 0.95, 95% CI: 0.87, 1.05) (Table 2). The control regression analysis, but not the

**Table 2. Effect of each software transition and any hardware transition on daily minutes of exercise (negative binomial models with a random intercept per participant) among participants with activity data for all 7 days before and all 7 days after transitioning between the indicated software versions or between two different hardware versions.**

| Transition | Overall | | Female participants | | Male participants | |
|---|---|---|---|---|---|---|
| | Unadjusted RR (95% CI) | Adjusted RR (95% CI) | Unadjusted RR (95% CI) | Adjusted RR (95% CI) | Unadjusted RR (95% CI) | Adjusted RR (95% CI) |
| **Analysis: Change in software or hardware (week before vs. week after transition)** | | | | | | |
| watchOS 5 to 6 | 1.01 (0.94, 1.09) | 1.01 (0.94, 1.08) | 1.02 (0.90, 1.16) | 1.02 (0.88, 1.17) | 1.01 (0.92, 1.10) | 1.00 (0.92, 1.10) |
| watchOS 6 to 7 | 0.92 (0.86, 0.99) | 0.93 (0.87, 0.99) | 0.94 (0.84, 1.09) | 0.95 (0.83, 1.08) | 0.92 (0.84, 0.98) | 0.92 (0.85, 0.99) |
| watchOS 7 to 8 | 1.13 (1.06, 1.19) | 1.13 (1.06, 1.19) | 1.15 (1.05, 1.26) | 1.15 (1.04, 1.27) | 1.11 (1.04, 1.19) | 1.12 (1.04, 1.20) |
| watchOS 8 to 9 | 0.91 (0.86, 0.97) | 0.91 (0.86, 0.97) | 0.95 (0.87, 1.05) | 0.95 (0.86, 1.04) | 0.89 (0.83, 0.96) | 0.89 (0.83, 0.96) |
| Hardware change | 1.00 (0.92, 1.08) | 1.00 (0.93, 1.08) | 1.11 (0.94, 1.32) | 1.12 (0.92, 1.36) | 0.97 (0.90, 1.06) | 0.97 (0.89, 1.05) |
| **Control: No change in software or hardware (two weeks before vs. week before transition)** | | | | | | |
| watchOS 5 to 6 | 0.99 (0.92, 1.07) | 1.00 (0.93, 1.08) | 0.98 (0.84, 1.15) | 0.99 (0.86, 1.16) | 1.00 (0.91, 1.09) | 1.00 (0.91, 1.10) |
| watchOS 6 to 7 | 1.10 (1.03, 1.18) | 1.10 (1.03, 1.18) | 1.11 (0.97, 1.27) | 1.11 (0.95, 1.28) | 1.10 (1.01, 1.18) | 1.09 (1.01, 1.18) |
| watchOS 7 to 8 | 1.00 (0.94, 1.06) | 1.00 (0.95, 1.05) | 1.04 (0.95, 1.14) | 1.04 (0.95, 1.14) | 0.97 (0.91, 1.04) | 0.97 (0.91, 1.05) |
| watchOS 8 to 9 | 1.04 (0.98, 1.11) | 1.04 (0.98, 1.11) | 1.03 (0.94, 1.14) | 1.03 (0.94, 1.14) | 1.05 (0.98, 1.12) | 1.05 (0.98, 1.13) |
| Hardware change | 1.01 (0.92, 1.09) | 1.01 (0.93, 1.11) | 0.96 (0.76, 1.14) | 0.96 (0.79, 1.19) | 1.02 (0.94, 1.12) | 1.02 (0.93, 1.12) |

descriptive analysis, showed a statistically significant increase in daily exercise minutes before the watchOS 6 to 7 transition, overall (unadjusted RR = 1.10, 95% CI: 1.03, 1.18) and among male participants (unadjusted RR = 1.10, 95% CI: 1.01, 1.18) (Table 2). Both control analyses showed null results for the watchOS 8 to 9 transition (Fig 2 Panel B; Table 2).

The effects of software upgrades were similar but attenuated in sensitivity analyses run on the full sample (S1 Fig; S3 Table; S4 Table).

Daily active calories were lower after transitioning from watchOS 6 to 7 and watchOS 8 to 9. The average difference between mean daily active calories in the week before and week after upgrading from either watchOS 6 to 7 or 8 to 9 was approximately -23 calories overall, and both transitions had a larger impact on male participants (Fig 3 Panel A; S2 Table). In negative binomial regression models, watchOS 7 and watchOS 9 were both associated with 4% fewer active calories than watchOS 6 and watchOS 8, respectively, overall and among male participants (Table 3). No change was observed in the control analysis comparing two weeks before and one week before the respective upgrades (Fig 3 Panel B; Table 3; S2 Table). No other software transitions were associated with a change in daily active calories (Fig 3 Panel A; Table 3; S2 Table). The effect of each software transition was similar in sensitivity analyses, though the watchOS 6 to 7 transition was no longer statistically significant (S2 Fig; S3 Table; S5 Table).

## Hardware version

Hardware upgrades were not associated changes in daily exercise minutes (Fig 2 Panel A; Table 2; S2 Table) or active calories (Fig 3 Panel A; Table 3; S2 Table) for either male or female

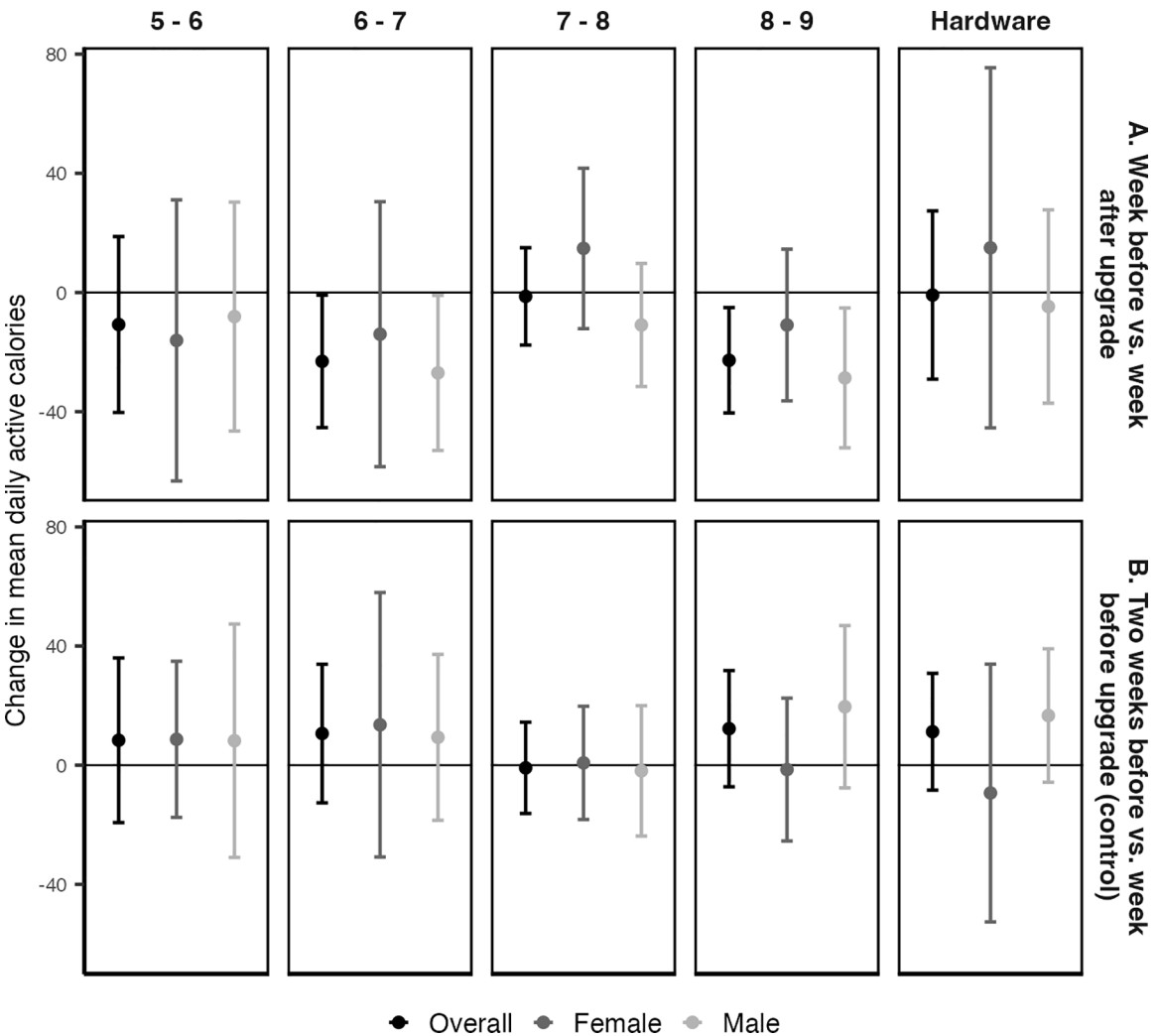

**Fig 3. Average difference in mean daily active calories in the week before and the week after transitioning between the indicated software versions.**

participants. Results were consistent across the primary and sensitivity analyses, with and without adjusting for day-of-the-week (S1 Fig; S2 Fig; S6 Table). The control analysis also showed no change in either outcome in the two weeks before a hardware upgrade (Fig 2 Panel B; Fig 3 Panel B; S3 Table; S6 Table).

## Discussion

We estimated large and statistically significant changes in daily exercise minutes and/or active calories measured by some, but not all, major software upgrades. The watchOS 7 to 8 transition was associated with an increase in daily exercise minutes but not active calories that was larger among female participants. Conversely, the watchOS 6 to 7 and 8 to 9 transitions were associated with decreases in both outcomes that were larger among male participants. Importantly, the relative change was larger and more clinically meaningful for exercise minutes.

These findings are unlikely to be fully explained by normal week-to-week variability or seasonality. First, most of the control analyses showed null results for both outcomes, suggesting

**Table 3. Effect of each software transition and any hardware transition on daily active calories (negative binomial models with a random intercept per participant) among participants with activity data for all 7 days before and all 7 days after transitioning between the indicated software versions, or between two different hardware versions.**

| Transition | Overall | | Female participants | | Male participants | |
|---|---|---|---|---|---|---|
| | Unadjusted RR (95% CI) | Adjusted RR (95% CI) | Unadjusted RR (95% CI) | Adjusted RR (95% CI) | Unadjusted RR (95% CI) | Adjusted RR (95% CI) |
| **Analysis: Change in software or hardware (week before vs. week after transition)** | | | | | | |
| watchOS 5 to 6 | 0.99 (0.96, 1.02) | 0.99 (0.96, 1.02) | 0.98 (0.94, 1.02) | 0.98 (0.94, 1.03) | 0.99 (0.95, 1.03) | 0.99 (0.95, 1.03) |
| watchOS 6 to 7 | 0.96 (0.94, 0.99) | 0.96 (0.94, 0.99) | 0.97 (0.92, 1.01) | 0.97 (0.92, 1.01) | 0.96 (0.93, 0.99) | 0.96 (0.93, 1.00) |
| watchOS 7 to 8 | 1.00 (0.98, 1.02) | 1.00 (0.98, 1.02) | 1.01 (0.98, 1.05) | 1.01 (0.98, 1.04) | 0.99 (0.96, 1.02) | 0.99 (0.96, 1.02) |
| watchOS 8 to 9 | 0.97 (0.94, 0.99) | 0.97 (0.95, 0.99) | 0.98 (0.95, 1.02) | 0.98 (0.95, 1.02) | 0.96 (0.93, 0.99) | 0.96 (0.93, 0.99) |
| Hardware change | 0.99 (0.96, 1.02) | 0.99 (0.96, 1.02) | 1.03 (0.95, 1.10) | 1.03 (0.96, 1.11) | 0.98 (0.95, 1.01) | 0.98 (0.94, 1.01) |
| **Control: No change in software or hardware (two weeks before vs. week before transition)** | | | | | | |
| watchOS 5 to 6 | 1.01 (0.98, 1.04) | 1.01 (0.98, 1.05) | 1.01 (0.96, 1.06) | 1.01 (0.96, 1.05) | 1.02 (0.98, 1.06) | 1.02 (0.98, 1.05) |
| watchOS 6 to 7 | 1.02 (0.99, 1.05) | 1.02 (0.99, 1.05) | 1.03 (0.98, 1.09) | 1.03 (0.98, 1.09) | 1.01 (0.97, 1.05) | 1.01 (0.98, 1.04) |
| watchOS 7 to 8 | 1.00 (0.98, 1.02) | 1.00 (0.98, 1.02) | 1.00 (0.96, 1.03) | 1.00 (0.96, 1.03) | 1.00 (0.97, 1.02) | 1.00 (0.97, 1.02) |
| watchOS 8 to 9 | 1.02 (1.00, 1.05) | 1.02 (0.99, 1.04) | 1.01 (0.97, 1.05) | 1.01 (0.97, 1.05) | 1.03 (0.99, 1.06) | 1.03 (0.99, 1.06) |
| Hardware change | 1.01 (0.98, 1.05) | 1.01 (0.97, 1.05) | 0.98 (0.91, 1.04) | 0.98 (0.92, 1.04) | 1.02 (0.98, 1.06) | 1.02 (0.98, 1.06) |

that physical activity is generally stable over a typical two-week period (Fig 2 Panel B, Table 2; Table 3, Fig 3 Panel B, S2 Table). However, for the watchOS 5 to 6 transition, a significant increase in exercise minutes during the control period suggests that unusually high activity levels in the week before the upgrade may partially account for the observed decrease following the transition. Future studies should consider evaluating software effects over longer timescales to account for short-term or unexpected fluctuations in activity levels. If the effects of a software transition were driven by seasonality, we would expect to see a similar effect in the control transition period, which overlapped with the actual transition period (Fig 1). Second, although half of the transitions occurred within September of the release year, participants transitioned on different dates, some at different times of year, and lived in various geographic regions. This makes it unlikely that the estimated effects of software transitions are confounded by seasonal or geographic factors, such as weather or temperature. Finally, if seasonality were a major factor, we would expect each software upgrade to show similar directional effects, given that they were all released in the fall of each year. This was not observed.

There are several potential explanations for the observed sex-differences in the magnitude and statistical significance of some transitions. First, the effect estimates are less precise for female participants due to smaller sample sizes. Second, activity algorithms may be sex-specific and updated independently. Third, the observed changes may be due to updates in how activity algorithms account for weight, which differs on average by sex. Finally, Apple Watch measures activity differently during user-initiated workouts (vs. the rest of the day), and across different workout types (e.g., indoor run vs. outdoor run vs. yoga) [20]. If a new software version were to adjust how activity is earned within workouts, or for specific workout

types, this could lead to sex-differences in exercise minutes and/or active calories if workout frequency and/or type varies between male and female users.

We found no change in either daily exercise minutes or daily active calories following hardware transitions. Importantly, this result represents the combined effect of switching between any two hardware versions. We could not isolate the change in activity measurement associated with transitioning to a specific hardware version, or between adjacent hardware versions. The effect of any individual hardware version is obscured due to pooling data from all possible transitions.

### Research implications

These results have important implications for physical activity research using consumer wearable devices. Most notably, failing to account for software version may introduce bias. Studies comparing physical activity levels between participants or over time, including analyses of longitudinal trends and/or the effect of an intervention, may be confounded by differences in software versions between people and/or over the course of the study period. A large majority of Apple Watch users in our sample installed all major software versions released during their study period. These upgrades are annual, so many software changes should be expected over the course of a multi-year longitudinal study. Based on our estimates, the confounding may be large and either over or underestimate the association of interest. As such, it is important that researchers using data from consumer wearables collect data about software and hardware versions when either varies between participants or over time.

### Limitations

The effect of individual hardware versions could not be assessed since few people transitioned between multiple and/or consecutive versions of hardware. We restricted data to 7 days before and after each transition and longer-term trends were not examined. We also did not consider the joint effects of software and hardware transitions, even though it was common for both upgrades to occur simultaneously.

Apple has undertaken several initiatives to help researchers leverage Apple Watch data for health-related studies. However, the lack of transparency regarding how these data are measured, processed, and summarized remain an important barrier to their use in research. Moving forward, researchers and wearable manufacturers need to collaborate to identify and address methodological challenges and barriers that currently limit the usability and impact of these datasets.

### Conclusions

Differences in software version over time or between people may confound physical activity analyses using Apple Watch data. Researchers should collect data about hardware and software and assess for risk of confounding by one or both factors.

### Supporting information

**S1 Table. Participant characteristics, overall and for each transition, among the full analytic sample (activity data for at least one day in both the pre- and post-transition week).**
(DOCX)

**S2 Table. Change in physical activity (a) during the transition period (week before vs. week after each transition), and (b) during the control transition period (two weeks before**

vs. one week before each transition), overall and stratified by gender, among the primary analytic sample (activity data for all 14 days of the respective period).
(DOCX)

**S3 Table. Change in physical activity (a) during the transition period (week before vs. week after each transition), and (b) during the control transition period (two weeks before vs. one week before each transition), overall and stratified by gender, among the full sample (activity data for all 14 days of the respective period).**
(DOCX)

**S4 Table. Effect of each software transition on daily minutes of exercise (negative binomial models with a random intercept per participant) among participants with activity data for at least one day in the 7 days before and 7 days after the specified software transition.**
(DOCX)

**S5 Table. Effect of each software transition on daily active calories (negative binomial models with a random intercept per participant) among participants with activity data for at least one day in the 7 days before and 7 days after the specified software transition.**
(DOCX)

**S6 Table. Effect of hardware transitions on daily exercise minutes and daily active calories (negative binomial models with a random intercept per participant) among participants with at least one day of activity data in the 7 days before and 7 days after a hardware transition.**
(DOCX)

**S1 Fig. Average difference in mean daily exercise minutes in the week before and after the specified software transition, using different minimum data thresholds.**
(TIFF)

**S2 Fig. Average difference in mean daily active calories in the week before and after the specified software transition, using different minimum data thresholds.**
(TIFF)

## Acknowledgments

The authors thank Mike Spears, Mobile Application Development (MAD) Lab Manager at the University of Toronto, for his expertise, flexibility, and support in developing the mobile application used to collect participant consent and data for this study. He was instrumental in the study's success and demonstrated the feasibility of this novel data collection approach within a university research setting and on a limited budget.

## Author contributions

**Conceptualization:** Shelby Lisabeth Sturrock, Rahim Moineddin, Dionne Gesink, Sarah Woodruff, Daniel Fuller.

**Data curation:** Shelby Lisabeth Sturrock, Daniel Fuller.

**Formal analysis:** Shelby Lisabeth Sturrock.

**Methodology:** Shelby Lisabeth Sturrock, Rahim Moineddin, Daniel Fuller.

**Supervision:** Rahim Moineddin, Dionne Gesink, Daniel Fuller.

**Writing – original draft:** Shelby Lisabeth Sturrock.

**Writing – review & editing:** Shelby Lisabeth Sturrock, Rahim Moineddin, Dionne Gesink, Sarah Woodruff, Daniel Fuller.

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
