## [Decision Letter · Decision Letter 0]

17 Sep 2024

PDIG-D-24-00257

The effect of software and hardware version on Apple Watch activity measurement: A secondary analysis of the COVFIT retrospective cohort study

PLOS Digital Health

Dear Dr. Shelby Lisabeth Sturrock,

Thank you for submitting your manuscript to PLOS Digital Health. After careful consideration, we feel that it has merit but does not fully meet PLOS Digital Health's publication criteria as it currently stands. Therefore, we invite you to submit a revised version of the manuscript that addresses the points raised during the review process.

Please submit your revised manuscript within 60 days Nov 16 2024 11:59PM. If you will need more time than this to complete your revisions, please reply to this message or contact the journal office at digitalhealth@plos.org. Please include the following items when submitting your revised manuscript:

We look forward to receiving your revised manuscript.

Kind regards,

Cleva Villanueva, M.D., Ph.D.

Guest Editor

PLOS Digital Health

Cleva Villanueva

Guest Editor

PLOS Digital Health

Additional Editor Comments (if provided):

The authors need to clearly state how the results could impact research, even if it seems obvious. If the research is conducted while the watch undergoes software or hardware updates, could this affect the results?

Exercise minutes and active calories are presented as percentages of change, but is it possible to show the mean and standard deviation (SD) of minutes and calories as well? This would help determine how active the cohort is. Additionally, is there a way to present the significance (p-value) of these changes?

Were all the data collected in September, when the software version was updated? Did the authors record data from two consecutive weeks with the same software and hardware versions? This would help clarify whether the observed changes are due to the software or hardware update, rather than normal week-to-week variations.

Are the differences between men and women influenced by variations in their weights and body sizes?

Was sleep time included in the total time the watch was worn?

The statistical analysis needs to be explained more clearly.

It is recommended to separate Table 1 into two tables, with one focusing on participant characteristics

Reviewers' comments:

Reviewer's Responses to Questions

**Comments to the Author**

1. Does this manuscript meet PLOS Digital Health’s publication criteria ? Is the manuscript technically sound, and do the data support the conclusions? The manuscript must describe methodologically and ethically rigorous research with conclusions that are appropriately drawn based on the data presented.

Reviewer #1: Yes

Reviewer #2: Yes

Reviewer #3: Yes

2. Has the statistical analysis been performed appropriately and rigorously?

Reviewer #1: Yes

Reviewer #2: Yes

Reviewer #3: Yes

3. Have the authors made all data underlying the findings in their manuscript fully available (please refer to the Data Availability Statement at the start of the manuscript PDF file)?

Reviewer #1: No

Reviewer #2: No

Reviewer #3: No

4. Is the manuscript presented in an intelligible fashion and written in standard English?

PLOS Digital Health does not copyedit accepted manuscripts, so the language in submitted articles must be clear, correct, and unambiguous. Any typographical or grammatical errors should be corrected at revision, so please note any specific errors here.

Reviewer #1: Yes

Reviewer #2: Yes

Reviewer #3: Yes

5. Review Comments to the Author

Please use the space provided to explain your answers to the questions above. You may also include additional comments for the author, including concerns about dual publication, research ethics, or publication ethics. (Please upload your review as an attachment if it exceeds 20,000 characters)

Reviewer #1: 1*. Does this manuscript meet PLOS Digital Health’s publication criteria? Is the manuscript technically sound, and do the data support the conclusions? The manuscript must describe methodologically and ethically rigorous research with conclusions that are appropriately drawn based on the data presented.

This manuscript does meet PLOS Digital Health’s publication criteria.

This manuscript is clearly written with sound design and provides useful information for researchers planning to use wearables. The conclusion that software updates can impact measurements is important to establish. 

*2. Has the statistical analysis been performed appropriately and rigorously?

The statistical analysis has been performed appropriately and rigorously.

As the software updates arrive in September each year and the activity trends go both up and down with software updates, time of year is somewhat accounted for. Line 169 does explain accounting for this but it is unclear if the 14 day window is fixed or not. Please clarify. It could be useful to analyze the same two weeks in September in the people who did not apply their software updates until November. In a similar way, an analysis of two weeks with the same software at any point could show what a normal deviation week to week might occur. This would provide more confidence when activity significantly changed for males but not females between updates if this trend is not seen when software updates stay the same. 

*3. Have the authors made all data underlying the findings in their manuscript fully available (please refer to the Data Availability Statement at the start of the manuscript PDF file)?

No but the authors state that it may be available on request – pending ethical approval.

*4. Is the manuscript presented in an intelligible fashion and written in standard English?

Yes , the manuscript is well written and clear.

*5. Review Comments to the Author

Additional comments 

Table 1 displays some general cohort characteristics. However, the mean activity levels and standard deviations for the cohorts analyzed are not shown. Although the changes in activity levels are shown as a percentage and reported in minutes, it would be interesting to understand how active the cohort is to understand how meaningful the % change or 5 minutes reported difference is.

The authors discuss how sex specific algorithms may mean different changes in men and women, but also might want to mention how different mean weights and sizes of males v females might make change more detectable in larger males. 

Line 159 – people are excluded if they wear their watch less than 10 hours per day – does this include wearing it when asleep or exclude sleep time?

Line 186 and Line 202– Clarifying the use of the terms the primary analysis and the descriptive analysis. Please can you clarify if the descriptive analysis is a secondary analysis that does not require data every day or if descriptive refers to the type of analysis.

Reviewer #2: This article studies an often neglected issue. However, comparing measurements between software versions does not seem to carry significant input to the research community. The key is still whether these measurements are reasonably accurate proxies to the 'real' data. While confounding of software versions may be an issue, is that a significant issue for the research outcome? Under what scenarios is the versioning impacting the research outcome?

Reviewer #3: Overall this is a nice article, well presented, and making an important contribution to the field. 

Introduction:

Well written, provides a good background to the problem. Reflections on existing literature may be a little bit light. 

Methods:

Well written, clearly presented. 

Using 40 minutes without an HR measurement to define non-wear time seems to depend on the algorithm Apple is using – and so potentially the software version? If overall results are impacted by non-wear time, could you be detecting an artefact where Apple is changing how it records heart rate but actually its actigraphy measurements remain the same? It is likely not having a big effect, but would be good to know if this has been considered. 

Results:

Table 1 is a little bit confusing. The first two sets of rows show outcomes 1 and 2, while the remainder show participant characteristics. I think it would be easier to interpret as two separate tables – with outcomes and participant characteristics separate. 

Discussion:

Overall conclusions are valid and research implications are important to the field. 

That software changes can affect results differentially depending on sex/gender is an important point. This could be reinforced by further statistical analysis comparing female vs male, I think at the moment it is just based on the averages being different.

6. PLOS authors have the option to publish the peer review history of their article (what does this mean? ). If published, this will include your full peer review and any attached files.

**Do you want your identity to be public for this peer review?** For information about this choice, including consent withdrawal, please see our Privacy Policy .

Reviewer #1: Yes: Dr Sarah Johnson

Reviewer #2: No

Reviewer #3: No

---

## [Decision Letter · Decision Letter 1]

17 Dec 2024

The effect of software and hardware version on Apple Watch activity measurement: A secondary analysis of the COVFIT retrospective cohort study

PDIG-D-24-00257R1

Dear Shelby Lisabeth Sturrock,

We are pleased to inform you that your manuscript 'The effect of software and hardware version on Apple Watch activity measurement: A secondary analysis of the COVFIT retrospective cohort study' has been provisionally accepted for publication in PLOS Digital Health.

Best regards,

Cleva Villanueva, M.D., Ph.D.

Guest Editor

PLOS Digital Health

**Additional Editor Comments (if provided):**

The authors addressed all the comments raised by the reviewers. It would be important exclude sleep time in future analysis

**Reviewer Comments (if any, and for reference):**

Reviewer's Responses to Questions

**Comments to the Author**

1. If the authors have adequately addressed your comments raised in a previous round of review and you feel that this manuscript is now acceptable for publication, you may indicate that here to bypass the “Comments to the Author” section, enter your conflict of interest statement in the “Confidential to Editor” section, and submit your "Accept" recommendation.

Reviewer #1: All comments have been addressed

2. Does this manuscript meet PLOS Digital Health’s publication criteria ? Is the manuscript technically sound, and do the data support the conclusions? The manuscript must describe methodologically and ethically rigorous research with conclusions that are appropriately drawn based on the data presented.

Reviewer #1: Yes

3. Has the statistical analysis been performed appropriately and rigorously?

Reviewer #1: Yes

4. Have the authors made all data underlying the findings in their manuscript fully available (please refer to the Data Availability Statement at the start of the manuscript PDF file)?

Reviewer #1: No

5. Is the manuscript presented in an intelligible fashion and written in standard English?

Reviewer #1: Yes

6. Review Comments to the Author

Reviewer #1: The author's have addressed all previous comments - the addition of a control analysis was particularly helpful. While this paper refers to Apple Watch data this is a useful and relevant paper for researcher's working in the wearables field where measurement changes due to software updates are likely overlooked.

The author's clarified that the minimum of 10 hours wear included sleep time. This means it is possible for participants to only wear the watch at night - this seems unlikely and the average activity time is high enough to suggest this isn't the case but future analysis could try removing sleep time from the analysis.

The data availability statement suggests the underlying data is not currently available publicly but could be on request.

7. PLOS authors have the option to publish the peer review history of their article (what does this mean? ). If published, this will include your full peer review and any attached files.

**Do you want your identity to be public for this peer review?** For information about this choice, including consent withdrawal, please see our Privacy Policy .

Reviewer #1: **Yes: ** Sarah Johnson
